# Prediction of Extraintestinal Manifestations in Inflammatory Bowel Disease Using Clinical and Genetic Variables with Machine Learning in a Latin IBD Group [note 1]

**DOI:** 10.3390/ijms26125741

**Published:** 2025-06-15

**Authors:** Tamara Pérez-Jeldres, Paula Reyes-Pérez, Patricio Gonzalez-Hormazabal, Cristóbal Avendano, Roberto Segovia Melero, Lorena Azocar, Veronica Silva, Andres De La Vega, Elizabeth Arriagada, Elisa Hernandez, Nataly Aguilar, Carolina Pavez-Ovalle, Cristian Hernández-Rocha, Roberto Candia, Juan Francisco Miquel, Manuel Alvarez-Lobos, Ivania Valdes, Alejandra Medina-Rivera, Maria Leonor Bustamante

**Affiliations:** 1Departmento de Gastroenterología, Pontificia University Católica de Chile, Santiago 8330024, Chile; rcasegovia@gmail.com (R.S.M.); lazocarlopez@gmail.com (L.A.); elisa.hernandez.c@gmail.com (E.H.); n.aguilar.estay@gmail.com (N.A.); cdpavez@yahoo.es (C.P.-O.); cristian.hernandez.rocha@gmail.com (C.H.-R.); roberto.candia@gmail.com (R.C.); jmiquelp@uc.cl (J.F.M.); manalvarezlobosl@gmail.com (M.A.-L.); 2Departmento de Gastroenterología, Hospital san Borja Arriaran, Santiago 8360160, Chile; veritosilvaf@gmail.com (V.S.); imnotep@gmail.com (A.D.L.V.); earriagadah@gmail.com (E.A.); 3Laboratorio Internacional de Investigación Sobre el Genoma Humano, University Nacional Autónoma de Mexico, Mexico 76230, Mexico; paularoxanarp@gmail.com (P.R.-P.); medina.alexiel@gmail.com (A.M.-R.); 4Programa de Genética Humana, Facultad de Medicina, Instituto de Ciencias Biomédicas (ICBM), Santiago 8380453, Chile; patriciogonzalez@uchile.cl; 5Departmento de Ciencias de la Computación, Pontificia University Católica de Chile, Santiago 7820436, Chile; coavendano@uc.cl; 6Departmento de Enfermedades Respiratorias, Escuela de Medicina, Pontificia University Católica de Chile, Santiago 8330024, Chile; ivaniavaldes.a@gmail.com

**Keywords:** inflammatory bowel disease, extraintestinal manifestation, genetic variants

## Abstract

Extraintestinal manifestations (EIMs) significantly increase morbidity in inflammatory bowel disease (IBD) patients. In this study, we examined clinical and genetic factors associated with EIMs in 414 Latin IBD patients, utilizing machine learning for predictive modeling. In our IBD group (314 ulcerative colitis (UC) and 100 Crohn’s disease (CD) patients), EIM presence was assessed. Clinical differences between patients with and without EIMs were analyzed using Chi-square and Mann–Whitney U tests. Based on the genetic data of 232 patients, we identified variants linked to EIMs, and the polygenic risk score (PRS) was calculated. A machine learning approach based on logistic regression (LR), random forest (RF), and gradient boosting (GB) models was employed for predicting EIMs. EIMs were present in 29% (120/414) of patients. EIM patients were older (52 vs. 45 years, *p* = 0.01) and were more likely to have a family history of IBD (*p* = 0.02) or use anti-TNF therapy (*p* = 0.01). EIMs were more common in patients with CD than in those with UC without reaching statistical significance (*p* = 0.06). Four genetic variants were associated with EIM risk (rs9936833, rs4410871, rs3132680, and rs3823417). While the PRS showed limited predictive power (AUC = 0.69), the LR, GB, and RF models demonstrated good predictive capabilities. Approximately one-third of IBD patients experienced EIMs. Significant risk factors included genetic variants, family history, age, and anti-TNF therapy, with predictive models effectively identifying EIM risk.

## 1. Introduction

Inflammatory bowel disease (IBD) is an immune-mediated disorder arising from the interaction of genetic, microbial, and environmental factors that trigger a dysregulated immune response [1]. IBD encompasses Crohn’s disease (CD) and ulcerative colitis (UC) and is considered a systemic condition with varying courses and progression [2]. While primarily affecting the gastrointestinal tract, IBD can also involve extraintestinal organs, with extraintestinal manifestations occurring in 6% to 47% of patients [3,4,5,6,7].

Extraintestinal manifestations (EIMs) can be classified into three categories: *classical*, including inflammatory processes occurring at distant sites (e.g., uveitis and pyoderma gangrenosum); *associations*, which refers to links with other immune-mediated disorders (e.g., myocarditis, multiple sclerosis, and interstitial lung disease); *complications*, which encompass the indirect effects of systemic inflammation and the side effects of treatment (e.g., anemia, venous thromboembolism, and pancreatitis by azathioprine). The pertinence of considering EIMs as complications is a topic of debate, as their causal relationship with the disease is mediated by additional factors [3]. On the other hand, *classical* and *associated* EIMs may share pathological mechanisms with IBD, although the precise mechanisms are only partially understood. Some hypotheses propose a role for cross-reactivity or molecular mimicry, aberrant lymphocyte homing, and a pro-inflammatory state characterized by the upregulation of cytokines and chemokines. These factors can lead to an imbalanced immune response [8]. Additionally, changes in the intestinal microbiota and genetic predisposition have been suggested as contributing elements [9,10,11].

A challenge in understanding EIMs in IBD is determining whether to approach them with a “lumping” or “splitting” strategy. In support of the “lumping” strategy, there are reasons to hypothesize that EIMs share common mechanisms, as is the case of autoimmune disease in general, warranting their conjoint analysis. For instance, recent research has highlighted novel pathogenetic mechanisms of IBD with EIM, emphasizing the role of the melanocortin system in disease inflammation. The melanocortin system constitutes a network of peptides, encompassing α-, β-, and γ-melanocyte-stimulating-Hormone (MSH), as well as adrenocorticotropic hormone (ACTH), which interacts with melanocortin receptors (MC1-5R) to regulate immune responses and inflammation [12]. Emerging evidence suggests that melanocortins play a crucial role in the pathophysiology of IBD, influencing both intestinal and systemic inflammation. Additionally, their involvement in conditions such as arthritis underscores their broader immunomodulatory potential [13]. These findings might support the hypothesis that dysregulation of melanocortin signaling may contribute to the persistence of inflammation beyond the gut, reinforcing the need for further investigation into targeted therapies that address both IBD and its associated EIMs [12,13]. On the other hand, this does not explain the apparent heterogeneity in their presentation or why some patients have a specific tissue affected. It is possible that each type of EIM has its own biological mechanisms, and thus, they must be studied separately. However, it is also possible that their differences are not due to causal but to modifying factors, in which case they can be analyzed as a whole.

Regarding genetic susceptibility to IBD, several genetic loci have been identified as risk factors. In this context, pleiotropy, the phenomenon where a single DNA variant influences multiple traits, may account for an overlapping genetic background and a shared pathophysiology between IBD and EIMs [14]. In support of this hypothesis, a strong familial concordance in EIMs has been observed [15,16].

Several genes have been associated with the EIMs pathogenesis, and differences in the prevalence of EIMs among populations with different ancestries support the concept of genetic susceptibility [4]. However, understanding them in the context of IBD remains challenging because of the small sample size of IBD subphenotypes, which limits statistical power. Khrom et al. [17] recently published the most extensive multicenter study on the pathogenesis of EIMs in the IBD context, analyzing 11,894 patients, with 1704 in the EIMs+ and 10,190 in the EIMs− group. This sample was mostly made up of White individuals (>95%), where only 2% approximately self-declared to be Hispanic. In the study, key factors associated with EIMs included female sex, smoking status, IBD subtype, surgery, colonic Crohn’s disease, positive anti-nuclear cytoplasmic antibody (ANCA), and positive anti-Saccharomyces cerevisiae (ASCA). Furthermore, associations with tumor necrosis factor, JAK-STAT, IL-6, major histocompatibility complex, and CPEB4 (cytoplasmic polyadenylation element-binding protein 4) genes were identified [17].

Previously thought to affect individuals of European ancestry primarily, IBD has become a global concern in recent years. Despite this, most genetic research has focused on Europeans, with limited studies on non-European populations, primarily in East and South Asia. Few susceptibility loci have been detected in these studies, likely due to small sample sizes [18]. Latin American countries, which have also seen increasing IBD incidence, have a heterogeneous ancestral composition, with varying proportions of European, African, and Native American genetic variants across countries [19]. Chile’s genetic admixture is also derived from Native Americans, Spaniards, enslaved Africans, and European migrants, with Chileans averaging 53% European and 42% Amerindian (18% Aymara and 25% Mapuche) [20]. The median global ancestry distribution of our studied IBD population is 58% European, 39% Amerindian, and 2% African [19,21,22]. Further research to understand genetic risk factors influencing IBD phenotypes in the Andean region of South America is needed. Expanding genetic research beyond European cohorts is essential for identifying specific susceptibility loci and enhancing predictive models for EIMs. Strengthening these efforts could improve polygenic risk scores and enable targeted interventions for diverse populations.

EIM of IBD presents a serious clinical challenge, impacting quality of life, increasing complications, and potentially raising morbidity and mortality [23]. The relationship of EIMs to the underlying disease is unclear, as they can appear both before and after the IBD diagnosis, and they can be either linked to or independent of disease activation [23]. EIMs affect multiple organs and sometimes require specialized treatment distinct from intestinal inflammation. They have a strong effect on clinical outcomes, worsening disease progression, predicting the need for biological therapy, and increasing the risk of surgery and IBD relapse [24]. Despite the growing recognition of EIMs as a key aspect of IBD, significant challenges in diagnosis, classification, and treatment persist, mainly due to inconsistencies in defining and identifying EIMs across clinical settings [25]. With the exponential growth of big data in healthcare, including genomics, transcriptomics, imaging, therapeutics, and electronic health records, machine learning (ML) models offer a promising solution to enhance EIM prediction and personalized treatment approaches [26]. Compared to traditional statistical methods, ML algorithms excel in capturing nonlinear relationships within complex datasets, leading to more precise risk stratification and treatment optimization. By integrating clinical and genetic variables, ML models can enhance risk assessment, facilitating early intervention and tailored treatment strategies that address both intestinal inflammation and systemic complications [26].

Recent advancements in ML applications for IBD have emphasized genetic and molecular profiling as key components in predicting EIM susceptibility. Supervised learning techniques, such as random forests, gradient-boosting machines (GBMs), and deep neural networks, have demonstrated significant potential in identifying genetic markers associated with disease severity and EIM risk [27]. Additionally, unsupervised learning approaches, including clustering algorithms and principal component analysis (PCA), have been utilized to uncover hidden patterns in genomic data, helping researchers stratify patients into risk categories. By leveraging these advanced methodologies, predictive models can guide therapeutic decisions, enhance precision medicine, and ultimately enhance long-term patient outcomes in IBD management [28]. A study of 152 patients with Crohn’s disease assessed the predictive accuracy of three classification techniques, naïve Bayes, Bayesian additive regression trees, and Bayesian networks, for identifying extraintestinal manifestations. Researchers examined three sets of variables: clinical characteristics, risk factors, and relevant genetic polymorphisms. Among the 75 patients who developed extraintestinal manifestations, Bayesian networks, a probabilistic graphical model used in machine learning to analyze uncertainty and dependencies between variables, demonstrated the highest accuracy, achieving 82% with clinical data alone and 89% when genetic information was incorporated. This study exemplifies how machine learning tools can enhance the prediction of extraintestinal manifestations in Crohn’s disease, enabling more accurate risk stratification and personalized strategies to improve patient care [28].

Our study sought to identify predictors of EIMs in Latin American individuals, focusing on the known clinical and genetic factors associated with EIMs in a Chilean IBD group from 2019 to 2024. We employed strategies to identify associations between clinical and genetic factors and EIMs. We designed predictive models for EIMs using machine learning techniques, and we developed polygenic risk scores.

## 2. Results

### 2.1. Clinical Results

A total of 414 patients were studied, and 120 (29%) reported one or more EIMs. The most common EIMs were arthritis/arthralgias/ankylosing spondylitis at 32% (38/120), skin disorders at 13% (16/120), and thrombotic events at 11% (13/120). Figure 1 and Table 1 detail EIM distribution and the clinical characteristics of the EIMs+ and EIMs− groups. In the EIMs+ group, 69% had UC, whereas the prevalence of the latter was 79% in EIMs− patients. In the CD group, 37% (37/100) were EIMs+, whereas 26% (83/314) of patients with UC were EIMs+, which did not reach significance (*p*-value = 0.06). The group with EIMs was significantly older than the group without EIMs (mean of 52 vs. 45 years, *p*-value = 0.001), used more anti-TNF drugs (18% vs. 9%, *p*-value = 0.01), and had a higher frequency of a positive family history of IBD (26% vs. 12%, *p*-value = 0.02). In a univariate regression model, a family history of IBD was significantly associated with EIMs+ status (odds ratio (OR) 2.47, confidence interval (CI) 1.44–4.23, *p*-value = 0.001), as well as the use of anti-TNF (OR 2.21, CI 1.20–4.08, *p*-value 0.01; Table 2). Although the medians of Body Mass Index (BMI, *p*-value = 0.006)), white blood cell count (WBC, *p* = 0.04), C-reactive protein (CRP, *p*-value = 0.01) levels, and erythrocyte sedimentation rates (ESRs, *p*-value = 0.01) were significantly higher in the EIMs+ group, the regression model did not show a significant association (BMI, *p*-value = 0.07, WBC, *p*-value = 0.306, CRP, *p*-value = 0.74, ESRs, *p*-value = 0.74).

### 2.2. Genotyping and Genetics Variants

In a subset of 232 IBD patients, we investigated 205 genetic variants previously identified as being associated with EIMs, according to Khrom et al. [16]. Chilean patients were initially genotyped using the Illumina Infinium array, examining 357,392 single-nucleotide polymorphisms (SNPs), including 49 of the 205 variants in this study. After data imputation, a total of 20,393,242 variants were identified, including 154 variants in this study (Appendix A). We then tested the association with EIMs in our sample using case–control, univariate, and multiple regression analyses adjusted for sex, age, and ancestry, employing additive, recessive, dominant, and three-genotype models. A total of four genetic variants were associated with the risk of EIMs, as shown in Table 3 and Table 4. The association with rs9936833-CC (LINC917 and FENDRR) was associated with EIMs (OR 2.92, CI 1.32–6.55, *p*-value = 0.008) in the univariate analysis and the multivariate analysis for the additive model (OR 3.09, CI 1.38–7.09, and *p*-value = 0.01). Other variants significantly associated with EIM risk were rs3132680 (TRIM31, TRIM31-AS1), rs3823417 (PSORS1C1), and rs4410871 (PTV1). Table 3 and Appendix A show the genetic variants associated with EIMs. Table 4 shows the gene location associated with the variants here found and previously reported by Khrom [17]. Table 5 shows the distribution of the risk allele of the SNP found to be associated with EIMs.

### 2.3. Polygenic Risk Score to Predict Extraintestinal Manifestation

To investigate the relationship between genetic risk scores and EIMs in the context of IBD, we constructed a PRS using the genetic variants and associated beta coefficients reported in Khrom’s study [17]. We calculated the Z-scores for a total of 205 patients (74 with extraintestinal manifestations, EIMs+, and 131 without, EIM−). Our analysis did not reveal any significant differences in the mean Z-scores between the two groups, as illustrated in Figure 2. The area under the receiver operating curve (AUROC) curve was 0.69, and the accuracy of the model was 66.34%, which represents the proportion of correct predictions out of the total cases. The balanced accuracy, which provides a more comprehensive assessment by considering both sensitivity and specificity, was 58%. This metric is particularly relevant for evaluating models in scenarios where class distributions are not equal.

Next, we repeated the same analysis using the genetic variants and beta coefficients reported in Liu’s study [18], which are associated not specifically with EIMs but with IBD risk. In the context of pleiotropism (where a gene may influence two or more seemingly unrelated phenotypic traits), it could be worthwhile to develop a PRS (Figure 3). However, we observed a similar pattern to that shown in Figure 2.

### 2.4. Machine Learning Models to Predict Extraintestinal Manifestation

#### 2.4.1. Logistic Regression for EIMs

We developed a logistic regression model using genetic variables and clinical factors to predict the occurrence of EIMs, calculating odds ratio and confidence intervals for the model coefficients. The variables included in the model were rs4410871, rs9936833, rs3132680, and rs3823417, along with clinical and demographic factors such as a family history of IBD, BMI, sex, current anti-TNF therapy, smoking status, disease duration, ESRs, European ancestry (EUR), African ancestry (AFR), Amerindian ancestry (AMR), CRP, age, white blood cell count, hemoglobin levels, platelet count, and albumin level. The data were analyzed globally and divided into training and test sets to evaluate the model’s ability to generalize results, indicating that the AUROC curve for the test data outperformed that of the PRS (AUROC LR total 0.78, train 0.79, and 0.88 test data vs. AUROC PRS 0.68). The AUROC and performance metrics for the model for predicting EIMs are presented in Figure 4 and Table 6.

Next, we evaluated the variables significantly associated with EIM risk in both the univariate and multivariate logistic regression models, as shown in Table 7.

A family history of IBD and the use of anti-TNF therapy were the most relevant clinical variables associated with EIMs. In contrast, the genotypes rs9936833-TT, rs3132680-CA, and rs382417-GA were found to be protective against EIMs.

#### 2.4.2. Random Forest and Gradient Boosting

We conducted a random forest and gradient-boosting analysis using the same data and variables as those in the logistic regression model. We performed 80/20 split data into new subsets for random forest training and testing. The objective was to assess the significance of these variables in predicting EIMs and to evaluate their predictive power within an alternative classification framework.

The confusion matrix in Figure 5 for the random forest models illustrates the precision, recall, and F1-score for both (a) the training dataset and (b) the testing dataset. The results highlighted a tendency for overfitting in the training data, as balanced data for each category label showed 119 instances of label 1 (EIMs+) and 119 instances of label 0 (EIMs−), achieving perfect accuracy (100%). The model’s performance was evaluated using several key metrics presented in a tabular format for the training and testing datasets: precision, recall, F1-score, and support. Precision measures the accuracy (proportion of correct predictions made by a classification model out of the total predictions), indicating the percentage of true positive results among all positive predictions. The recall represents the model’s ability to identify all relevant instances, specifically the proportion of actual positives that were correctly classified. The F1-score balances precision and recall, representing a harmonic mean of the two metrics and offering a single score that captures both properties. Support refers to the number of actual occurrences of each class in the dataset.

As shown in Figure 5, for the training data, the model achieved a precision of 100%, a recall of 100%, and an F1-score of 100%, with a support of 119 instances for the positive class (EIMs+) and the same number for the negative class (EIMs−). Conversely, on the test data, the precision was 66%, the recall stood at 77%, and the F1-score reached 71%, with a support of 30 instances for the positive class. These metrics highlight the model’s robustness and ability to classify the target variable and demonstrate reasonable generalization ability on the testing dataset, achieving an overall accuracy of 0.68 and an AUROC of 0.79.

Additionally, we evaluated clinical and genetic factors predicting EIMs using a gradient-boosting model as shown in Figure 6. On the test dataset, the model showed an F1-score of 74%, with an accuracy of 72%, indicating reasonable generalization. Despite perfect metrics on the training data revealing overfitting, the model obtained an AUC for the tested data of 78%.

## 3. Discussion

This study aims to characterize EIMs in Chilean patients with IBD by assessing the clinical and genetic factors relevant to this population. Through the application of various predictive models, these factors were validated to enhance their validity (rs3132680, rs3823417, rs4410871, rs9936833, IBD family history, use of anti-TNF, and age). Precise identification of risk factors might enable a more effective approach to detecting patients predisposed to EIMs, facilitating earlier diagnosis and timely therapeutic interventions in clinical practice [30].

Twenty-nine percent of this Latin IBD group reported EIMs, which is in accordance with the range previously described. The prevalence and incidence of EIMs depend on the types of EIMs included in the definition, with variable prevalence rates as mentioned above [23].

In the Khrom et al. [17] study, the evaluated EIMs included ankylosing spondylitis and sacroiliitis (AS-SI), pyoderma gangrenosum (PG), erythema nodosum (EN), psoriasis (PS), ocular inflammation, primary sclerosing cholangitis (PSC), and peripheral arthritis (PA). They found that 27% of the 12,083 patients with IBD had at least one EIM. PA was reported by 17% (n = 2040) of patients, 6.5% (n = 783) reported skin disorders, 3.8% (n = 459) reported ocular manifestations, and 2.4% (n = 286) reported other EIMs. In our studied IBD group, PA was present in 25% (n = 31) of patients, and skin disorders (including PG, EN, PS, atopic dermatitis, and pemphigus) were reported by 13% (n = 16), while seven patients (5%) reported ocular manifestations. These findings highlight the similarities and differences in the prevalence of EIMs between Khrom et al.’s study and our IBD group, underscoring the importance of considering population-specific factors in the characterization of EIMs in IBD.

Significant differences were observed in EIM prevalence between Caribbean countries (Cuba, Puerto Rico, and the Dominican Republic) and Latin American countries (Colombia, Ecuador, Mexico, Peru, Uruguay, and Venezuela), with rates of 59.1% and 29.5%, respectively (*p*-value 0.001, OR 3.44, CI 2.84–4.18) [31]. Similar trends have been observed in Western countries compared with Eastern countries, with EIMs being more frequent in the West, particularly primary sclerosing cholangitis [32,33]. Furthermore, articular involvement occurs more often in African American and Asian individuals, and EIMs are more frequently diagnosed in Indian subjects compared with Malay or Chinese individuals [34,35].

Our research study aimed to identify genetic and clinical predictors of EIMs. We used candidate SNPs previously nominated by Khrom et al. [17], but our definition of EIMs was broader. Not only do we include complications as well as classical manifestations and associations, but we also do not limit our analysis to the specific associations previously identified. We used this approach as we hypothesized that shared pathophysiological mechanisms between IBD and EIMs may not be specific to a certain type of EIMs [16,36,37]. While this approach provides stronger results and broader conclusions, it can oversimplify the unique differences among EIMs, making detailed analysis more difficult. Nonetheless, this method helps to uncover general patterns and highlights how clinical and genetic factors influence EIMs in individuals with IBD. However, we acknowledge that a limitation of our study is the classification of diverse EIMs as a single group, which may obscure unique patterns among specific subphenotypes. Additionally, the small sample sizes for individual EIM-IBD subphenotypes restrict our statistical power, making it difficult to draw definitive conclusions regarding their genetic associations.

In our study, TNF therapies were significantly associated with EIM risk. Due to their role in modulating systemic inflammatory responses, anti-TNF therapies can address the heightened inflammatory tone associated with IBD, which increases the risk of EIM development through shared genetic and environmental factors [37]. Elevated inflammatory mediators, such as TNF, contribute to intestinal inflammation and activate immune responses at non-intestinal sites, leading to EIMs. Anti-TNF therapies have demonstrated reasonable response rates for treating cutaneous manifestations, arthritis, and ocular EIMs in IBD, suggesting a role for TNF-dependent mechanisms in EIM pathophysiology. However, these drugs can also induce skin lesions, contributing to the burden of skin disease in IBD. The mechanisms behind these lesions are unclear but may involve cytokine imbalances and immune cell distribution changes due to TNF inhibition. Increased IFNα release can cause psoriasis, while TNF blockade might decrease Th1 and Th17 cell accumulation at inflammation sites, causing compensatory expansion elsewhere [37].

The median age was significantly higher in the extraintestinal manifestation group, 52 years, compared with 45 years in the other group (*p*-value 0.001). In univariate regression, the OR for age was 1.01 (CI 1.00–1.04, *p*-value 0.02), indicating that each additional year increases the event probability by 1%. In multivariate regression, the OR rose to 1.04 (CI 1.01–1.06, *p*-value 0.006), suggesting that after adjusting for other variables, each additional year is linked to a 4% increase in event probability. These results suggest that increasing age might be a significant risk factor for developing EIMs in this Latin group. The increase in risk with age may relate to extended exposure to the inflammatory process, extrinsic factors such as diet or medication, or the effect of immunosenescence. This is an interesting topic for future analysis.

A family history of IBD was significantly associated with 28% of patients showing extraintestinal manifestations (compared with 12% of patients EIMs−, odds ratio [OR] 2.47, confidence interval [CI] 1.44–4.23, *p*-value 0.001) in our studied IBD group. Similarly, other studies have identified a family history of IBD as an independent predictor of the condition [16]. Additionally, the contribution of genetic risk to the pathogenesis of EIMs is reflected in the concordance rates of 70–80% among parent–child pairs or siblings with IBD [15]. Notably, differences in EIM prevalence among ethnicities support the notion of genetic background in IBD or family-common shared environmental factors that increase susceptibility to the disease.

In contrast to the Khrom study [17], we did not identify female sex, smoking, surgery, and the CD colonic phenotype as risk factors for EIMs, but this could be due to our study’s small sample size.

Multiple genetic variants have been identified with an increased risk of EIM pathogenesis. In our study, the SNPs rs9936833, rs3132680, rs3823417, and rs4410871 were associated with the risk of EIMs. The rs9936833-C allele was associated with high risk in the different tested models. The SNPs in Khrom’s study [17] were related to the presence of any of the following diseases: AS-SI, EN, PG, Ps, EYE, PSC, or PA.

These SNPs map to the *LINC00917* (long intergenic non-protein-coding RNA 917) and *FENDRR* (*FOXF1* adjacent non-coding developmental regulatory RNA) genes. *LINC00917* has been linked to other diseases, including colorectal cancer (CRC), facilitating CRC proliferation, metastasis, and aerobic glycolysis through c-Myc, suggesting its potential as a therapeutic target for CRC treatment [38]. Additionally, *LINC00917* has been associated with CpG sites and is suggested to be a potential biomarker for detecting gestational diabetes (GDM) in women [39]. This is intriguing because SNPs located at CpG sites can influence the interaction between the genome and the epigenome in GDM, and maybe *LINC00917* could have a similar role to EIMs related to IBD [40]. *FENDRR* is also a long, non-coding RNA that regulates the IFNγ-induced M1 phenotype in macrophages [41]. Thus, genetic variants, particularly SNPs linked to *LINC00917* and *FENDRR*, may play a role in EIM development, suggesting a connection between genetic regulation and disease pathogenesis. Further research is needed to elucidate these relationships and their potential implications.

Interestingly, for the rs3132680, rs3823417, and rs4410871 variants, the heterozygous genotype was associated with an increased risk of EIMs. rs3132680 maps to the *TRIM31* and *TRIM-AS* genes.

The tripartite motif (TRIM) family includes over 70 proteins that function as E3 ubiquitin ligases and regulate key cellular processes, playing a role in diseases such as IBD. In DSS-induced murine models, *TRIM31* was found to enhance the ubiquitin–proteasome pathway of the NLRP3 inflammasome in macrophages, which are linked to IBD pathogenesis, and it downregulates intestinal inflammation by inducing autophagy in CD [42]. Given the potential role of *TRIM31*-related variants in modulating inflammatory pathways in IBD, further research could explore their influence on EIM risk and disease progression.

The rs3823417 maps to the psoriasis susceptibility 1 *(PSORS1)* locus, which is located within the major histocompatibility complex and is one of the main genetic determinants for psoriasis [43], which was one of the EIMs reported in our studied IBD group. The rs4410871 variant is harbored in the Plasmocytoma Variant Translocation 1 *(PVT1)* gene, a long non-coding RNA found on chromosome 8q24. *PVT1* is upregulated in UC and is associated with the activation of inflammatory cytokines and pathways such as MAPK/NF-kB. Moreover, *PVT1* holds promise as a diagnostic marker for UC for distinguishing patients from healthy controls [44]. In the Khrom study [17], it was associated with PSC.

We admit the limitation of our genetic association analysis is the sample size, which affects statistical power and the ability to detect significant variants. While the Bonferroni correction was applied, no associations remained significant under this method. However, the genetic variants analyzed in our study have been previously reported as significant in larger IBD cohorts, supporting their potential relevance. These findings emphasize the importance of further validation in larger, independent populations to strengthen the understanding of genetic associations in diverse patient groups.

The PRS exhibits a lower predictive capability to predict EIMs than the logistic regression and RF and GB models. This discrepancy may be attributed to the PRS relying solely on the additive effects of multiple genetic variants, which can overlook complex interactions between predictors and crucial factors. Furthermore, if genes do have a pleiotropic effect, they might play a non-specific role in increasing the risk of several related phenotypes. Then, the specific pathology would be determined by gene–environment interactions, which would not be observed in differences in the PRS, considering that it does not include the effect of other predictors different from SNPs. On the other hand, important genes may be relevant to this Chilean IBD group that are not included among the characterized SNPs based on Liu [18] and Khrom’s studies [17]. Liu and Krohm’s studies focus on European populations and may not capture key genetic variants relevant to Latin populations. Since no IBD GWAS studies have been conducted in Chilean or Latin American populations, key genetic variants influencing disease susceptibility may remain undiscovered and are, therefore, not accounted for in the PRS. This highlights the need for population-specific genetic studies to improve prediction accuracy.

We recognize the inherent challenges in the transferability of PRS when applied to the Chilean population, given that the ethnicity studied differs from that of the reference GWAS. This limitation arises from the absence of genetic studies specific to IBD in Latin American populations. Without a reference dataset tailored to this ancestry, the beta coefficients required for precise PRS calculations are lacking, leading to potential inaccuracies in risk estimation. While cross-ancestry PRS models have demonstrated varying degrees of applicability, studies indicate that European-derived PRSs often lose predictive power in admixed populations, including those of Latin American descent [45]. The absence of recalibration within our study underscores the broader need for dedicated genomic research within this population to enhance PRS accuracy and clinical utility. Future research efforts should focus on refining the applicability of PRS in diverse populations by integrating admixture-aware statistical approaches, leveraging multi-ancestry meta-analyses, and conducting local genome-wide studies to mitigate transferability biases and enhance predictive performance.

In contrast to PRS, the machine learning models used can incorporate relevant clinical, demographic, and genetic variables, thereby enhancing their predictive accuracy. Furthermore, while the PRS assumes additive effects, logistic regression can account for non-additive effects among variables. This flexibility enables ML models to better capture the heterogeneity of complex diseases, ultimately yielding a more precise risk assessment. These models confirmed the significance of clinical factors such as a family history of IBD, age, and anti-TNF therapy, alongside the genotypes rs9936833, rs3132680, rs3823417, and rs4410871, in predicting EIMs.

Advances in machine learning and artificial intelligence (AI) have significantly improved the prediction and analysis of EIM and IBD. A logistic regression model achieved 96% accuracy in predicting IBD-associated arthropathy, paving the way for early diagnosis [46]. An NLP system outperformed traditional coding methods in detecting EIMs from clinical notes [25]. AI-based clustering revealed unique disease communities and interaction networks, enhancing our understanding of EIM prevalence [47]. However, Bayesian machine learning techniques showed less accuracy than classical models, underscoring the need for improved methodologies [28]. Our study integrates genetic variants with clinical and demographic factors for enhanced predictive precision. Unlike population-level clustering, we focus on individualized risk assessment, integrating genetic and clinical data to improve EIM predictions in our local cohort.

Our predictive models demonstrated good capabilities for identifying EIMs. Specifically, the logistic regression model achieved an AUROC of 0.89 on the test set, indicating effective differentiation between the two groups. While the random forest model showed a tendency for overfitting in the training data, it still demonstrated reasonable generalization ability on the testing dataset, achieving an overall accuracy of 0.68 and an ROC curve score of 0.79. The gradient-boosting model also showed a reasonable generalization with an F1-score of 0.74, an accuracy of 0.72 on the test dataset, and an AUC of 0.78.

Importantly, the fact that all three distinct models (logistic regression, random forest, and gradient boosting) yielded predictive capabilities validates that the selected clinical and genetic variables were useful in predicting EIMs in our IBD group. It is relevant to note that the random forest and gradient-boosting models used a dataset that employed oversampling techniques, which allowed for a balanced dataset to train the models.

## 4. Materials and Methods

### 4.1. Study Population

We conducted a transversal study at Hospital San Borja Arriarán (HSBA), a tertiary referral center for IBD in Chile, from January 2020 to July 2024. Patients were invited to participate if they had received a diagnosis of IBD at least 3 months prior supported by clinical, endoscopic, histologic, and imaging findings in accordance with clinical guidelines and International Disease Classification criteria [48,49,50]. Demographic data and relevant clinical variables were recorded, including age, sex, family history, and smoking status (see Table 1). The presence (EIMs+) or absence (EIMs−) of any of the following EIMs was recorded from the clinical chart: peripheral arthritis, ankylosing spondylitis, pyoderma gangrenosum, erythema nodosum [3], atopic dermatitis [51], pemphigus [52], thrombotic events [53], stroke or cardiovascular disorders [54], ocular manifestations (episcleritis/uveitis) [11], urolithiasis [55], autoimmune pancreatitis [56], pulmonary manifestations of IBD [57], sensitive-motor polyneuropathy [58], and other unspecified conditions (patients with EIMs noted in their clinical records without specific diagnoses documented for those conditions). This study included patients from comparable socioeconomic backgrounds, classified as working class (D) and lower-middle class (C3), according to the scale established by the Association of Market Researchers and Public Opinion in Chile [59]. Ethics approval was obtained from the Institutional Review Boards of the Servicio de Salud Metropolitano Central/HSBA, Santiago, Chile (IRB: 43/2022, Fondecyt Initiation grant number 11220147), and all participating patients provided written informed consent. Patients with missing or insufficient demographic or clinical information were excluded. Specifically, cases where variables had more than 30% missing values were not considered for analysis.

### 4.2. Genotyping and Genetic Risk Score Calculation

Blood samples in aliquots of 5 mL were collected from 232 participants and stored in vacutainer tubes with EDTA. DNA was extracted using the Invisorb Blood Universal kit (Invitek, Berlin, Germany; # ref 1031150200), following the manufacturer’s guidelines. The samples were stored at −80 °C and subsequently genotyped at Erasmus MC in the Netherlands using Illumina’s Infinium Global Screening Array to analyze 725,497 SNPs. Genotype quality control (QC) was conducted using R Studio, version 4.2.2, with the plinkQC library.

The perMarkerQC function was used to assess missingness rates across samples, deviations from Hardy–Weinberg Equilibrium (HWE), and minor allele frequencies (MAFs), using a threshold of 0.01. Additionally, the perIndividualQC function was used to evaluate total heterozygosity rates, missingness, the concordance of the assigned sex with the SNP sex, relatedness to other individuals in the study, and the genetic ancestry of the samples in the PLINK dataset. The genetic data were prepared using PLINK1, PLINK2, and Python 3.11. Post-quality control datasets were checked for strand flips using a strand correction tool as described in https://github.com/HALFpipe/ImputationProtocol (accessed on 9 June 2025). The corrected files were converted to PLINK2 format and subsequently to VCF files, which were divided by chromosome. Imputation was performed using the TOPMed v3 reference panel (build: hg38), with Eagle for phasing, a population-inclusive approach, and an r^2^ filter of 0.3. More information about the Imputation Server [60] and Minimac Imputation [61] can be found in their corresponding publications. Following imputation, the output files were merged and converted back to PLINK format, and the coordinates were updated to hg37 using liftOver. SNP-specific data were extracted, and the results were formatted for downstream analyses, including PRS calculations. The PRS was computed in PLINK, merging the score profile with covariate data. Scores were normalized to z-scores, and logistic regression was employed to analyze the PRS associations with disease status, adjusting for covariates. The PRS was calculated as the sum of the effect size (βi) of each genetic variant, obtained GWAs [17,18], multiplied by the individual’s genotype (Gi), which indicates the number of risk alleles present (coded as 0, 1, or 2).

The PRS was computed for 205 patients with complete data from principal component analysis (PCA). This approach captures key genetic variation and accounts for population structure and ancestry, ensuring robust analysis by adjusting for confounding effects associated with genetic background. Data visualizations included violin and quantile plots, with quantile groups for additional regression analysis. An ROC curve and confusion matrix assessed predictive accuracy, supported by density plots of predicted disease probabilities. To evaluate differences in the median Z-scores between groups with and without EIM, we performed a Welch two-sample *t*-test for both polygenic risk score sets. The results were summarized in tables and figures, with visual outputs saved for reporting.

We utilized R libraries for genetic data processing (genio version 1.0.25, plinkFile version 0.21), data manipulation (tidyverse version 2.0.0, dplyr version 1.1.4, data.table version 1.15.4, readr version 2.1.5), visualization (ggplot2 version 3.5.1, plotROC version 2.3.1), machine learning (caret version 6.0), and data management (openxlsx version 4.2.5.2).

To estimate ancestry, we performed an analysis using ADMIXTURE [62] with a reference panel from the 1000 Genome Project and HapMap, including 43 Native Americans, 56 Europeans, and 55 Africans, along with 342 Chileans. The 43 Native American samples demonstrated 99% or higher Native American ancestry, drawn from populations such as Nahua, Maya, Quechua, and Aymara [22].

### 4.3. Data Analysis

#### 4.3.1. Clinical Predictors

We assessed the prevalence of EIMs and analyzed clinical differences among groups with and without EIMs using Chi-square tests for categorical variables and the Mann–Whitney U test for median comparisons. For the clinical variables that were significantly different in both groups (age, family history of IBD, BMI, WBC, ESRs, CRP, and anti-TNF use), we carried out a univariate regression analysis to determine their association.

To ensure data quality and minimize the impact of missing values on our analysis, we applied a predefined exclusion criterion. Clinical variables with more than 20% missing values were removed from the dataset. For the remaining variables, we employed appropriate imputation techniques: mean imputation was applied to continuous variables, while mode imputation was used for categorical variables.

#### 4.3.2. Genetic Predictors

Additionally, we evaluated associations with the EIM susceptibility variants reported in Khrom’s study [17] in 232 IBD genotyped patients, using case–control univariate and multiple regression analyses under additive, recessive, dominant, and three-genotype models.

#### 4.3.3. Machine Learning Models

##### Regression Model

To explore the relationship between the clinical and genetic variables related to EIMs, a logistic regression model was developed, incorporating predictors such as the genetic variants associated with EIMs in our IBD group (rs4410871, rs9936833, rs3132680, and rs3823417), a family history of IBD, BMI, sex, anti-TNF therapy status, smoking, disease duration, ESRs, ethnicity, CRP, age, white blood cell count, hemoglobin, and platelets. Coefficients, ORs, and confidence intervals were calculated, with predictions being evaluated by employing confusion matrices for sensitivity and specificity across various cutoff probabilities (0.2, 0.5, and 0.7). The dataset was divided into training and test sets to assess model performance (70/30 split of 232 IBD patients with 83 EIMs+ and 149 EIMs−), using ROC curves to determine the AUC. Univariate analyses were also performed to clarify associations with EIMs, and all results were documented for interpretation.

##### Random Forest and Gradient Boosting

The methodology for developing the two classifiers, random forest and gradient boosting, followed a systematic series of steps aimed at optimizing predictive performance for a binary classification task. The implementation was supported by several powerful Python libraries.

Pandas version 2.23 was used for data manipulation and analysis, providing seamless handling of datasets through DataFrames. NumPy version 2.20 facilitated efficient numerical computations, particularly for array operations and mathematical functions. For machine learning, Scikit-learn version 1.6.0 served as the backbone, offering a wide range of algorithms and tools, including the random forest and gradient-boosting classifiers, alongside utilities for model evaluation and validation. The Imbalanced-learn library was utilized to apply the synthetic minority over-sampling technique (SMOTE), addressing class imbalance by generating synthetic instances of the minority class. This was carried out to balance the data for the random forest and gradient-boosting models. Balancing is crucial in machine learning when dealing with datasets where one class is significantly underrepresented compared to others. Without balance, the model may be biased towards the majority class, leading to poor predictive performance for the minority class. Additionally, Matplotlib version 3.10.0 and Seaborn version 0.13.2 Python libraries were employed for data visualization, offering an enhanced understanding of model performance through informative plots, such as confusion matrices.

Random forest is an ensemble learning method that constructs multiple decision trees during training and outputs the mode of their predictions. This approach helps mitigate overfitting and enhances accuracy. In contrast, gradient boosting is a sequential ensemble method that builds decision trees iteratively, with each new tree correcting the errors made by the previous ones. This allows for a more adaptable and precise model.

The dataset underwent initial preprocessing, which included an 80/20 split for training and testing. Its features were standardized using StandardScaler to ensure all variables contributed equally to the model’s performance. Stratified sampling was employed to divide the data into training and testing sets, preserving class proportions. Hyperparameter tuning for the random forest classifier was performed using GridSearchCV, which evaluated various parameter combinations, such as the number of estimators, maximum depth, and minimum sample sizes, across five-fold cross-validation. The best-performing model was selected, and its effectiveness was assessed using metrics such as recall, F1-score, and ROC AUC.

A similar process was applied to the gradient-boosting classifier, with hyperparameters tailored to its specific requirements. Lastly, a comparison of the performance of both models was conducted, providing valuable insights into their effectiveness in accurately classifying the target variable.

## 5. Conclusions

In conclusion, in our Latino IBD group, one-third of patients with IBD reported experiencing EIMs. In our study, we not only identified four genetic variants previously associated with EIMs in predominantly Caucasian populations but also highlighted the significant clinical connections between age and a family history of IBD as important risk factors. Patients with EIMs were more likely to receive anti-TNF therapies, supporting a relationship between inflammation and the development of EIMs. These findings are consistent with the existing literature. Furthermore, our machine learning predictive approach showcased superior predictive capability compared with the PRS. As ML continues to evolve, integrating diverse genetic backgrounds into predictive models will improve healthcare equity and ensure that genetic risk assessments are tailored to underrepresented populations. Expanding the genetic characterization of non-European cohorts could refine prediction models, ultimately facilitating more targeted interventions and optimized therapeutic strategies for IBD patients worldwide. By incorporating diverse populations, research can address disparities in treatment responsiveness and improve healthcare accessibility for underrepresented groups.

## Figures and Tables

**Figure 1 ijms-26-05741-f001:**
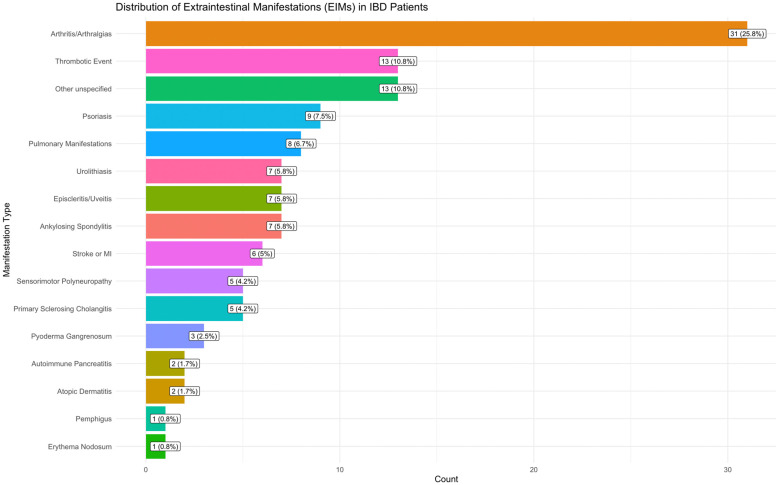
Frequency of extraintestinales manifestation in the Chilean IBD group.

**Figure 2 ijms-26-05741-f002:**
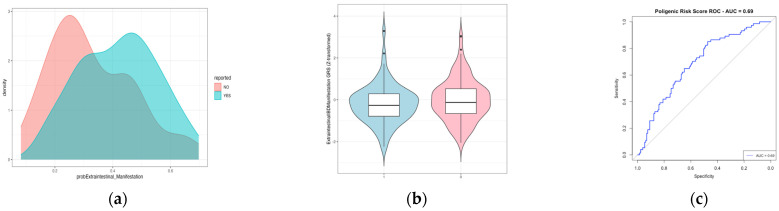
The polygenic risk score for extraintestinal manifestations based on the data reported in Khrom’s study. (**a**) The density plot illustrates the overlap in predicting extraintestinal manifestations (EIMs). (**b**) The box plot illustrates the Z-scores for each group (0 = EIM− and 1 = EIM+), showing no significant difference between their means. The Welch two-sample t-test results indicate a test statistic of t = 1.3925, degrees of freedom = 160.8, and *p*-value = 0.1657, confirming the absence of statistical significance. (**c**) The AUROC analysis is presented. EIM: extraintestinal manifestation.

**Figure 3 ijms-26-05741-f003:**
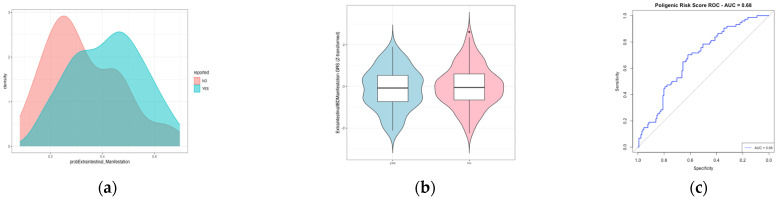
The polygenic risk score for extraintestinal manifestations based on the data reported in Liu’s study. (**a**) The density plot illustrates the overlap in predicting extraintestinal manifestations (EIMs). (**b**) The box plot illustrates the Z-scores for each group (0 = EIM− and 1 = EIM+), showing no significant difference between their means. The Welch two-sample t-test results indicate a test statistic of t = 0.79421, degrees of freedom = 153.47, and *p*-value = 0.4283, confirming the absence of statistical significance. (**c**) The AUROC (area under receiver operating curve) analysis is presented.

**Figure 4 ijms-26-05741-f004:**
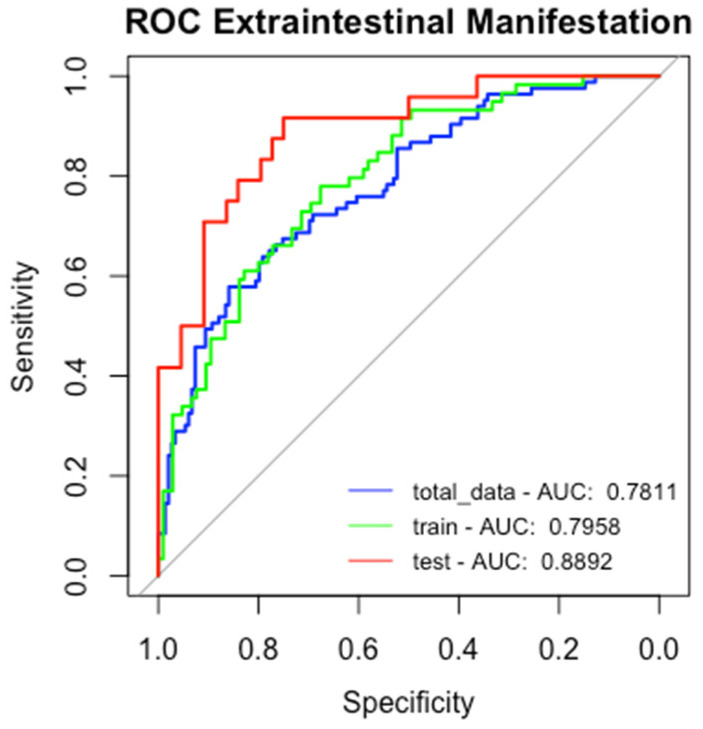
ROC of logistic regression model.

**Figure 5 ijms-26-05741-f005:**
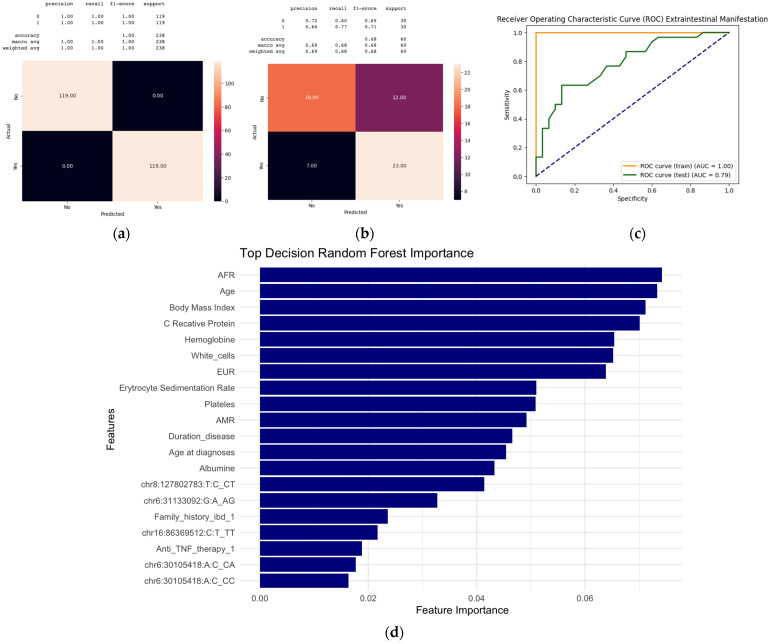
Performance metrics and feature importance in random forest classification for predicting extraintestinal manifestations. (**a**) The confusion matrix of the training data shows perfect performance: 119 true negatives, 119 true positives, and no errors, with perfect metrics (1.00) for both classes. (**b**) The confusion matrix of the test data shows 18 true negatives, 23 true positives, 12 false positives, and 7 false negatives. The classification report provides the following metrics: Class 0 (EIMs−): precision, 0.72; recall, 0.60; F1-score, 0.65. Class 1(EIMs+): precision; 0.66; recall, 0.77; F1-score, 0.71. The overall accuracy is 0.68. Precision = true positives/(true positives + false positives), recall = true positives/(true positives + false negatives), F1-score = 2 × (precision × recall)/(precision + recall), and overall accuracy = (true positives + true negatives)/total predictions. AUC = area under curve. (**c**) The AUROC plot illustrates the model’s sensitivity and specificity for both the training (AUC = 1.00) and test datasets (AUC = 0.79). (**d**) This bar chart displays the top features influencing the random forest model’s decisions. Feature importance variables for EIM prediction. chr16:86369512 = rs9936833, chr8:127802783 = rs4410871, chr6:31133092 = rs3823417, and chr6:30105418 = rs3132680. EIM: extraintestinal manifestation; ROC: receiver operating characteristic; AUC: area under curve.

**Figure 6 ijms-26-05741-f006:**
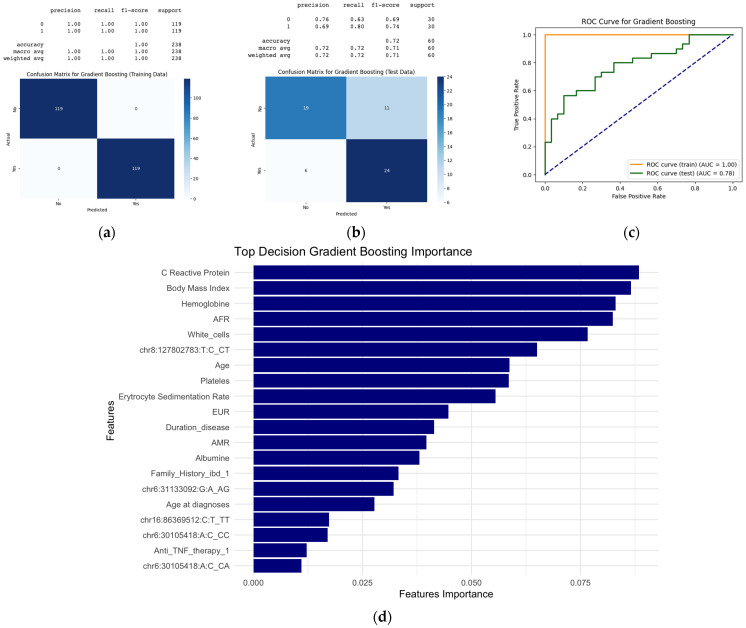
Performance metrics and feature importance in the gradient-boosting model for predicting extraintestinal manifestations. (**a**) The confusion matrix of the training data shows perfect performance: 119 true negatives, 119 true positives, and no errors, with perfect metrics (1.00) for both classes. (**b**) The confusion matrix of the test data shows 18 true negatives, 23 true positives, 12 false positives, and 7 false negatives. The classification report provides the following metrics: Class 0 (EIM−): precision, 0.72; recall, 0.60; F1-score, 0.65. Class 1(EIM+): precision; 0.66; recall, 0.77; F1-score, 0.71. The overall accuracy is 0.68. Precision = true positives/(true positives + false positives), recall = true positives/(true positives + false negatives), F1-score = 2 × (precision × recall)/(precision + recall), and overall accuracy = (true positives + true negatives)/total predictions. (**c**) The AUROC plot illustrates the model’s sensitivity and specificity for both the training (AUC = 1.00) and test datasets (AUC = 0.79). AUC = area under curve. (**d**) Feature importance variables for EIM prediction. chr16:86369512 = rs9936833, chr8:127802783 = rs4410871, chr6:31133092 = rs3823417, and chr6:30105418 = rs3132680. EIM: extraintestinal manifestation; ROC: receiver operating characteristic; AUC: area under curve.

**Table 1 ijms-26-05741-t001:** Clinical Characterization of IBD patients according to the presence of EIMs.

N = 414 (100%)	EIMs− (N = 294)	EIMs+ (N = 120)	*p*-Value
Ulcerative Colitis/Crohn’s	231 (78.6%)/63 (21.4%)	83 (69%)/37 (30%)	0.06
Age (years) (median, max–min)	45 (80–14)	52 (81–20)	**0.001**
Family history IBD (no/yes/NA)	256 (87%)/36 (12%)/2 (1%)	89 (74%)/31 (26%)	**0.02**
Sex (female/male)	168 (57%)/126 (43%)	80 (67%)/40 (37%)	0.09
Resective Surgery (no/yes)	266 (90%)/28 (10%)	103 (86%)/17 (14%)	0.22
Smoking (no/yes)	250 (85%)/44 (15%)	98 (82%)/22 (18%)	0.48
Body Mass Index (BMI, kg/m^2^)	26 (43–16)	27 (45–18)	**0.006**
Age Diagnoses (years) (median, min–max)	35 (77–4)	37 (67–7)	0.21
Hemoglobin (g/dL) (median, min–max)	14 (17–6)	14 (16.5–4)	0.31
White Cells (×10^9^/L) (median, min–max)	6.800 (2.550–19.200)	7.260 (3.500–14.900)	**0.04**
Platelets (μ/L) (median, min–max)	296.000 (17.600–743.000)	294,000 (67.000–616.000)	0.98
C Reactive Protein (mg/dL) (median, min–max)	0.6 (0.02–19.5)	0.6 (0.04–11.7)	**0.01**
Erythrocyte Sedimental Rate (mm/h) (median, min–max)	9 (0.5–125)	10 (0.2–72)	**0.01**
Albumine (median, max–min)	4.4 (5.23–2.0)	4.4 (5.2–3.4)	0.7
Steroids (no/yes)	252 (86%)/42 (14%)	99 (82.5%)/21 (17.5%)	0.49
Anti-TNF (no/yes)	267 (91%)/27 (9%)	98 (82%)/22 (18%)	**0.01**
Immunomodulator (no/yes/NA)	227 (77.2%)/66 (22.4%)/1 (0.4%)	86 (72%)/34 (28%)	0.52
Aminosalycilates (no/yes)	64 (22%)/230 (88%)	30 (33.3%)/90 (66.7%)	0.56
JAK inhibitors (no/yes)	292 (99.3%)/2 (0.7%)	120 (100%)/0	0.9
Anti-IL12/IL23 (no/yes)	291 (98.9%)/3 (1.1%)	119 (99%)/1 (1%)	1
Montreal Classification			
E1/E2/E3/NA	49 (21%)/60 (26%)/113 (50%)/9 (3%)	9 (11%)//23 (28%)/48 (58%)/3 (3%)	0.20
Montreal Classification CD			
Age			
A1/A2/A3/No registered	6 (9%)/37 (59%)/20 (32%)	2 (5%)/18 (49%)/17 (46%)	0.33
Localization			
Ileal (L1)	8 (13%)	7 (19%)	0.70
Colonic (L2)	35 (55%)	19 (51%)	
Ileocolonic (L3)	20 (32%)	11 (30%)	
Upper compromise (L4) (no/yes)	58 (92%)/5 (8%)	34 (92%)/3 (8%)	1
Behavior			
B1 (inflammatory)	39 (62%)	20 (54%)	0.74
B2 (structuring)	11 (17%)	8 (22%)	
B3 (penetrating)	13 (20%)	9 (24%)	
Perianal Disease (no/yes)	45 (71%)/18 (29%)	20 (54%)/17 (46%)	0.12

EIM: extraintestinal manifestation. *p*-value < 0.05 is highlighted in bold.

**Table 2 ijms-26-05741-t002:** Association of family history of IBD and anti-TNF usage with extraintestinal manifestations in patients.

Clinical Variable	Extraintestinal Manifestation Group				
Family History IBD	EIM − (n = 292)	EIM + (n = 120)	OR	CI	*p*-value *
No	256 (87%)	89 (74%)	Reference	Reference	0.001
Yes	36 (12%)	31 (28%)	2.47	1.44–4.23	
Anti-TNF	No (n = 294)	Yes (n = 120)	OR	CI	*p*-value
No	267 (91%)	98 (82%)	Reference	Reference	0.01
Yes	27 (9%)	22 (18%)	2.21	1.20–4.08	

* Chi-square, CI = confidence interval, and OR = odds ratio.

**Table 3 ijms-26-05741-t003:** Genotypic frequencies for rs9936833 in individuals according to the presence of extraintestinal manifestation.

rs9936833			Univariate			Adjusted by Ancestry, Sex, and Age			
Recesive	No = 149	Yes = 83	OR	CI	*p* -value	OR	Recesive	No = 149	*p*-value
TT_TC	128	61	Ref.	Ref.		Ref.	TT_TC	128	Ref.
CC	21	22	2.20	1.12–4.32	0.02	2.21	CC	21	0.02
Dominant	No = 149	Yes = 83	OR	CI	*p* -value	OR	Dominant	No = 149	*p*-value
TT	53	19	Ref.	Ref.		Ref.	TT	53	Ref.
CC_TC	96	64	1.85	1.02–3.49	0.05	2.00	CC_TC	96	0.03
Three Genotypes	No = 149	Yes = 83	OR	CI	*p* -value	OR	Three Genotypes	No = 149	*p*-value
CC	21	22	Ref.	Ref.	Ref.	Ref.	CC	21	Ref.
CT	75	42	0.53	0.26–1.08	0.08	0.55	CT	75	0.10
TT	53	19	0.34	0.15–0.75	0.008	0.32	TT	53	0.01
Additive	No = 149	Yes = 83	OR	CI	*p* -value	OR	Aditivo	No = 149	*p*-value
TT_0	53	19	Ref.	Ref.	Ref.	Ref.	TT_0	53	Ref.
TC_1	75	42	1.56	0.82–3.02	0.18	1.69	TC_1	75	0.12
CC_2	21	22	2.92	1.32–6.55	0.008	3.09	CC_2	21	0.01

Group classification (‘Yes’ or ‘No’) is based on EIM presence. OR: odds ratio; CI: confidence interval; Ref.: reference.

**Table 4 ijms-26-05741-t004:** Gene variants associated with estraintestinal manifestations in this study *.

CHR	SNP ^#^	Position-hg19 (Mb)	Candidate Gene (s)	Effect Allele	*p*-Value	Beta	EIM *
6	rs3132680	30,073,195	*TRIM31*, *TRIM31-AS1*	A	1,67 × 10 ^−5^	0.602	PS
6	rs3823417	31,100,869	*PSORS1C1*	A	5,79 × 10 ^−5^	0.351	PS
8	rs4410871	128,815,029	*PVT1*	A	7,51 × 10 ^−5^	−0.318	PSC
16	rs9936833	86,403,118	*LINC917*, *FENDRR*	G	7,68 × 10 ^−5^	−0.163	EIM7

CHR: chromosome; EIM: extraintestinal manifestation; SNP: single-nucleotide polymorphism; EIM7: subjects with evidence of any of the following: peripheral arthritis (PA); ankylosing spondylitis–sacroiliitis, pyoderma gangrenoum, erythema nodosus, psoriasis (PS), ocular manifestation, and primary sclerosing cholangitis (PSC). * Reported by Khrom et al. [16]. ^#^ Table 4 reports the chromosomal positions of SNPs, their associated candidate genes, effect alleles, GWAS-derived *p*-values, and beta coefficients, along with the corresponding EIMs, as reported in the Krohm study [16].

**Table 5 ijms-26-05741-t005:** Comparison population distribution risk allele associated with EIM in the IBD Chilean group *.

SNP	Alleles	IBD	IBD EIM+	IBD MEI-	Latin Population *	European Population *
rs3132680	A/C	0.295/ 0.705	0.295/ 0.705	0.295/ 0.705	0.242/ 0.758	0.320/ 0.680
rs3823417	A/G	0.261/ 0.739	0.240/ 0.760	0.272/ 0.728	0.195/ 0.805	0.242/ 0.758
rs4410871	T/C	0.338/ 0.662	0.361/ 0.639	0.326/ 0.674	0.401/ 0.599	0.307/ 0.693
rs9936833	C/T	0.416/ 0.583	0.518/ 0.481	0.392/ 0.607	0.427 /0.573	0.356/ 0.644

SNP frequency allele for the Latin and European populations was obtained from Ensembl. * [29] SNP: single-nucleotide polymorphism; IBD: inflammatory bowel disease; EIM: extraintestinal manifestation.

**Table 6 ijms-26-05741-t006:** Performance metrics of the EIM logistic regression model.

Dataset	Threshold	Accuracy	Sensitivity	Specificity	F1-Score
Full	0.20	0.60	0.88	0.44	0.61
Data	0.50	0.75	0.54	0.86	0.62
	0.70	0.71	0.24	0.97	0.36
Training	0.20	0.64	0.93	0.48	0.65
	0.50	0.75	0.59	0.84	0.63
	0.70	0.74	0.32	0.97	0.46
Testing	0.20	0.72	0.92	0.61	0.70
	0.50	0.84	0.66	0.91	0.72
	0.70	0.78	0.58	0.95	0.70

**Table 7 ijms-26-05741-t007:** Risk factors for extraintestinal manifestation according to logistic regression model.

	Univariate Analysis			Multivariate Analysis		
Variable	OR	CI	*p*-Value	OR	CI	*p*-Value
Family History of IBD	3.16	1.43–6.95	0.004	2.84	1.13–7.14	0.03
Age	1.02	1.00–1.04	0.02	1.04	1.01–1.06	0.007
Use Anti-TNF	2.94	1.32–6.51	0.007	5.10	1.82–14.27	0.002
rs9936833TT	0.34	0.15–0.75	0.008	0.21	0.08–0.57	0.002
rs3132680CA	0.29	0.12–0.72	0.007	0.24	0.08–0.72	0.01
rs3823417AG	0.30	0.15–0.75	0.008	0.25	0.07–0.85	0.03

OR: odds ratio; CI: confidence interval; IBD: inflammatory bowel disease.

## Data Availability

This study was approved by the Ethics Committee, which stipulates that the authors are not allowed to share the raw data in public repositories. However, data sharing is permitted through academic collaborations. Researchers interested in accessing the data can contact the corresponding authors directly under a reasonable request.

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
