# Peer review of "Prediction of Extraintestinal Manifestations in Inflammatory Bowel Disease Using Clinical and Genetic Variables with Machine Learning in a Latin IBD Group [Author-notes fn1-ijms-26-05741]"

_ijms, 2025, doi:10.3390/ijms26125741_

Round 1
Reviewer 1 Report
Comments and Suggestions for Authors
Dear Authors,
Very good article. Nonetheless, before to accept, due to the great work in machine learning and statistics that you have done, I encourage you to share with the readers the R code used through GitHub or other platforms.
Author Response
Dear Editors and Reviewers,
We sincerely appreciate your thoughtful reviews and valuable suggestions regarding our manuscript, Prediction of Extraintestinal Manifestations in Inflammatory Bowel Disease Using Clinical and Genetic Variables with Machine Learning in a Latin IBD Group. Your feedback has been instrumental in enhancing the clarity and rigor of our work.
We have carefully reviewed each of your comments and made the necessary adjustments. In the attached document, we provide a detailed response to every point raised, indicating the modifications made to the text.
We hope these changes adequately reflect your suggestions and enhance the manuscript's strength. Once again, we are grateful for your time and dedication in the review process, and we remain available for any further observations.
Sincerely,
Dra. Tamara Perez Jeldres
Reviewer 1
Very good article. Nonetheless, before to accept, due to the great work in machine learning and statistics that you have done, I encourage you to share with the readers the R code used through GitHub or other platforms.
Thank you for your kind words and valuable feedback. We truly appreciate your encouragement and recognition of the machine learning and statistical work we have done.
We agree with your suggestion and will make the code available to share.

Reviewer 2 Report
Comments and Suggestions for Authors
The study investigated the potential beneficial of machine learning to predict extraintestinal manifestations (EIMs) of inflammatory bowel disease (IBD) using clinical and genetic variables in a Latino cohort. Authors should take the following points into consideration when reviewing the manuscript:
- The introduction section focused on explaining what EIMs are, but did not explain why they are a real problem that warrants study, nor why machine learning models are being developed, as this is the core concept of the study. The value behind identifying EIM risks must also be clarified.
- In the introduction section, it is also good to mention any machine learning techniques, for modeling the role of genetic factors in EIM prediction, mentioned in previous studies.
- Please check the abbreviation (MEI) in Table 2 and add the full term for the abbreviations (OR, CI) below the table.
- under Table 4, "EIM7: peripheral arthritis (PA); ankylosing spondilitis–sacroilitis, pyoderma gangrenoum, erythema nodosus, psoriasis (PS), ocular manifestation, and primary sclerosing cholangitis (PSC). * as reported by Khrom et al. [16]", EIM7??. The sentence is confusing, are these genetic variants identified in the current study, or do they belong to reference 16?
- Regarding to "Polygenic Risk Score to predict Extraintestinal Manifestation", it's unclear how the results of other studies are relied upon. Please clarify.
- Below Figures 2 and 3, what does (T-test = 0.16) mean? Does it mean p, or is the t-test statistic used? Please also explain in the Materials and Methods section.
- The discussion section seems weak and needs more in-depth discussion. Furthermore, there are similar studies that have not been compared, so more effort in the discussion section would have made it stronger.
- The basis and goal of studying EIMs and their risks is to find an appropriate mechanism for treating them. So how can they be studied as a single group?
- Line 702, "ultimately facilitating more targeted interventions and optimized therapeutic strategies for IBD patients worldwide", This statement is inaccurate because treatments for inflammatory bowel disease are not necessarily a cure for EIMs, and vice versa. This is one of the real problems of EIMs, which requires a combination of treatments and interventions tailored to each EIM individually.
- "Prediction of extraintestinal manifestations in inflammatory bowel disease using clinical and genetic variables with machine learning, January 2025, Journal of Crohn s and Colitis 19(Supplement_1):i2352-i2354. DOI: 10.1093/ecco-jcc/jjae190.1479", is the abstract on ResearchGate related to the same study or not?
Author Response
Dear Editors and Reviewers,
We sincerely appreciate your thoughtful reviews and valuable suggestions regarding our manuscript, Prediction of Extraintestinal Manifestations in Inflammatory Bowel Disease Using Clinical and Genetic Variables with Machine Learning in a Latin IBD Group. Your feedback has been instrumental in enhancing the clarity and rigor of our work.
We have carefully reviewed each of your comments and made the necessary adjustments. In the attached document, we provide a detailed response to every point raised, indicating the modifications made to the text.
We hope these changes adequately reflect your suggestions and enhance the manuscript's strength. Once again, we are grateful for your time and dedication in the review process, and we remain available for any further observations.
Sincerely,
Dra. Tamara Perez Jeldres
Reviewer 2
The study investigated the potential beneficial of machine learning to predict extraintestinal manifestations (EIMs) of inflammatory bowel disease (IBD) using clinical and genetic variables in a Latino cohort. Authors should take the following points into consideration when reviewing the manuscript:
- The introduction section focused on explaining what EIMs are, but did not explain why they are a real problem that warrants study, nor why machine learning models are being developed, as this is the core concept of the study. The value behind identifying EIM risks must also be clarified.
Thank you very much for your valuable insights. We have incorporated this point into the introduction, specifically within the section between lines 120-136.
- In the introduction section, it is also good to mention any machine learning techniques, for modeling the role of genetic factors in EIM prediction, mentioned in previous studies.
Thank you very much for your valuable insights. We have incorporated this point into the introduction, specifically within the section between lines 139 and 158.
- Please check the abbreviation (MEI) in Table 2 and add the full term for the abbreviations (OR, CI) below the table.
Thank you for your feedback. We have reviewed Table 2 and confirmed the abbreviation (MEI). Additionally, we have added the full terms for OR (Odds Ratio) and CI (Confidence Interval) below the table as requested.
- under Table 4, "EIM7: peripheral arthritis (PA); ankylosing spondilitis–sacroilitis, pyoderma gangrenoum, erythema nodosus, psoriasis (PS), ocular manifestation, and primary sclerosing cholangitis (PSC). * as reported by Khrom et al. [16]", EIM7??. The sentence is confusing, are these genetic variants identified in the current study, or do they belong to reference 16?
EIM7 refers to subjects with evidence of any of the following peripheral arthritis (PA): ankylosing spondylitis–sacroiliitis, pyoderma gangrenous, erythema nodosus, psoriasis (PS), ocular manifestation, and primary sclerosing cholangitis (PSC) * as reported by Khrom et al.[16]. We added this explanation below the table.
- Regarding to "Polygenic Risk Score to predict Extraintestinal Manifestation", it's unclear how the results of other studies are relied upon. Please clarify.
Thank you for your comment. In our paper, we assess the predictive ability of genetic variants associated with EIM in IBD. Additionally, we performed the same analysis using genetic variants linked to IBD risk, both of which exhibited limited predictive capability. Since genome-wide association studies (GWAS) have not been conducted in Latin American populations, this analysis was intended as an exploratory approach.
- Below Figures 2 and 3, what does (T-test = 0.16) mean? Does it mean p, or is the t-test statistic used? Please also explain in the Materials and Methods section.
Thanks again by the suggestions, we change this legend for:
Figure 2. The box plot illustrates the Z-scores for each group (0 = EIM- and 1 = EIM+), showing no significant difference between their means. The Welch Two Sample t-test results indicate a test statistic of t = 1.3925, degree of freedom = 160.8, and a p-value = 0.1657, confirming the absence of statistical significance
Figure 3. The box plot illustrates the Z-scores for each group (0 = EIM- and 1 = EIM+), showing no significant difference between their means. The Welch Two Sample t-test results indicate a test statistic of t = 0.79421, degree of freedom = 153.47, and a p-value = 0.4283, confirming the absence of statistical significance.
We added in the methods: To evaluate differences in the median Z-scores between groups with and without EIM, we performed a Welch Two Sample t-test for both polygenic risk score sets. Lines 683-684.
- The discussion section seems weak and needs more in-depth discussion. Furthermore, there are similar studies that have not been compared, so more effort in the discussion section would have made it stronger.
We appreciate your feedback on strengthening the discussion. In response, we have expanded the section by comparing our findings with prior studies that utilized AI and machine learning for EIM prediction in IBD. Specifically, we highlight how our study integrates genetic variants alongside clinical factors, setting it apart from previous approaches. Additionally, we have incorporated reviewer comments to provide a more in-depth analysis of our results and their implications. These revisions enhance the discussion, making it more comprehensive and impactful.
- The basis and goal of studying EIMs and their risks is to find an appropriate mechanism for treating them. So how can they be studied as a single group?
Thank you very much for your thoughtful comment. As mentioned in the introduction, our study aims to identify potential predictors rather than imply a specific mechanism for each extraintestinal manifestation (EIM). We appreciate your insight on this matter.
Additionally, while a separate analysis could have been valuable, the reduction in sample size made this approach unfeasible. We have addressed this point in the discussion section to clarify our reasoning further.
Our research study aimed to identify genetic and clinical predictors of EIMs. We used candidate SNPs previously nominated by Khrom et al. [16], but our definition of EIMs was broader. Not only do we include complications as well as classical manifestations and associations, but we also do not limit our analysis to the specific associations previously identified. We used this approach as we hypothesized that shared pathophysiological mechanisms between IBD and EIMs may not be specific to a certain type of EIMs [14], [35], [36].
Lines 460-464 However, we acknowledge that a limitation of our study is the classification of diverse EIMs as a single group, which may obscure unique patterns among specific subpheno-types. Additionally, the small sample sizes for individual EIM-IBD subphenotypes restrict our statistical power, making it difficult to draw definitive conclusions regarding their genetic associations.
- Line 702, "ultimately facilitating more targeted interventions and optimized therapeutic strategies for IBD patients worldwide", This statement is inaccurate because treatments for inflammatory bowel disease are not necessarily a cure for EIMs, and vice versa. This is one of the real problems of EIMs, which requires a combination of treatments and interventions tailored to each EIM individually.
Thank you for your insightful feedback.
The statement in Line 702 does not explicitly refer to extraintestinal manifestations (EIMs) but instead emphasizes the broader need to characterize underrepresented populations in inflammatory bowel disease (IBD) research. Understanding genetic and clinical variability among minority groups is crucial, as studies in other conditions, such as lupus, have shown that treatment responses can differ based on ancestral background. While our paper does not delve into this aspect in detail, it is essential to emphasize the importance of inclusivity in research, particularly for populations with limited access to healthcare and research opportunities. We add to the paragraph
“Expanding the genetic characterization of non-European cohorts could refine prediction models, ultimately facilitating more targeted interventions and optimized therapeutic strategies for IBD patients worldwide”. By incorporating diverse populations, research can address disparities in treatment responsiveness and improve healthcare accessibility for underrepresented groups. Lines 776-778
- "Prediction of extraintestinal manifestations in inflammatory bowel disease using clinical and genetic variables with machine learning, January 2025, Journal of Crohn s and Colitis 19(Supplement_1):i2352-i2354. DOI: 10.1093/ecco-jcc/jjae190.1479", is the abstract on ResearchGate related to the same study or not?
Thank you for your inquiry. Yes, the abstract you mentioned is related to the same study. We had the opportunity to present this work at ECCO Berlin 2025 and DDW 2025, where it received positive feedback.

Reviewer 3 Report
Comments and Suggestions for Authors
I have reviewed with interest the original manuscript entitled “Prediction of extraintestinal manifestations in Inflammatory Bowel Disease using Clinical and Genetic variables with Machine Learning in a Latin Cohort” authored by Perez-Jeldres and colleagues.
The authors selected a cohort of IBD patients, with a marked predominance of UC cases (approximately 3:1 ratio compared to CD), in order to assess the potential occurrence of EIMs. They then analysed genetic data from approximately 232 patients within the overall group to identify genetic variants associated with EIMs, aiming to generate a polygenic risk score using a machine learning approach to predict which individuals would eventually develop EIMs (observed in roughly 30% of patients).
I have several major considerations:
- The study is cross-sectional in nature, making the term “cohort” somewhat inappropriate. This study design allows for the identification of associations, but it does not support establishing causality or true prospective prediction of EIMs.
- In my opinion, two biases are present: a selection bias (likely favouring more complex patients) and a representativeness bias (a geographically limited population with very limited generalisability, particularly in the context of genetic findings).
- No imputation of missing data has been performed, despite certain variables exhibiting over 30% missingness.
- The genetic association tests do not include any correction for multiple comparisons (e.g. Bonferroni). Given that more than one hundred tests are conducted, statistical significance would only be reached for p-values in the order of 10-4, meaning that some results with p-values merely below 0.05/0.01 may not actually be really significant.
- The ethnicity studied differs from that of the reference GWAS used for the polygenic risk score, which introduces a transferability bias due to the lack of recalibration of the score for the Chilean sample under investigation.
Minor considerations:
- I recommend introducing in the introduction section some novel pathogenetic mechanisms linking IBD with EIMs. There is emerging evidence regarding the role of the melanocortin system in IBD (https://pubmed.ncbi.nlm.nih.gov/38577176/), which is also implicated in conditions such as arthritis (https://pubmed.ncbi.nlm.nih.gov/37325641/). A brief mention of both studies as relevant references would be appropriate.
- The quality of Figure 1 is poor; it would be advisable to recreate it using more specialised software (e.g. GraphPad). Additionally, some of the pie chart segments at the bottom (e.g. purple and brown) overlap, obscuring the percentage labels.
- In Figures 5 and 6, the use of variable names as they appear in the statistical software is aesthetically unappealing. Labels such as “esr” or “age_diagnoses” should be replaced with properly formatted variable names.
Finally, I believe that greater emphasis should be placed, within the discussion, on the translational potential of these findings, on the research pathway that must continue to be pursued in this field, and on what is still lacking for the clinical integration of machine learning-based predictability of EIMs.
Despite these issues, some findings retain a degree of novelty. However, the major limitations of the study lie in its mathematical and methodological approach. These should either be addressed if possible, or clearly and explicitly stated in the limitations section, with a careful recalibration of the strength of the conclusions accordingly.
Comments on the Quality of English LanguageThe manuscript should be carefully reviewed for formatting and English syntax in order to improve readability and expository fluency.
Author Response
Dear Editors and Reviewers,
We sincerely appreciate your thoughtful reviews and valuable suggestions regarding our manuscript, Prediction of Extraintestinal Manifestations in Inflammatory Bowel Disease Using Clinical and Genetic Variables with Machine Learning in a Latin IBD Group. Your feedback has been instrumental in enhancing the clarity and rigor of our work.
We have carefully reviewed each of your comments and made the necessary adjustments. In the attached document, we provide a detailed response to every point raised, indicating the modifications made to the text.
We hope these changes adequately reflect your suggestions and enhance the manuscript's strength. Once again, we are grateful for your time and dedication in the review process, and we remain available for any further observations.
Sincerely,
Dra. Tamara Perez Jeldres
Reviewer 3
I have several major considerations:
- The study is cross-sectional in nature, making the term “cohort” somewhat inappropriate. This study design allows for the identification of associations, but it does not support establishing causality or true prospective prediction of EIMs.
Thank you for your insightful feedback. Even though this group of patients is actually part of a years-long follow-up study led by the first author since 2011, as the present manuscript does not report longitudinal data we have replaced the term cohort with the more accurate “individuals with IBD.
- In my opinion, two biases are present: a selection bias (likely favouring more complex patients) and a representativeness bias (a geographically limited population with very limited generalisability, particularly in the context of genetic findings).
Thank you for your thoughtful observation regarding potential biases. Regarding selection bias, the patients enrolled in this study at HSBA represent individuals primarily from lower and middle socioeconomic strata that belong to a central zone of Santiago. Their inclusion was not based on the complexity of their condition but rather on their accessibility to healthcare services. As demonstrated in Table 1, most patients in our study did not require major surgical interventions (89% of the total population did not undergo resective surgery) and did not need anti-TNF therapy (88% did not receive anti-TNF treatment). These findings indicate that the study population is not skewed toward more severe cases.
Additionally, in Chile, anti-TNF therapy is provided to all eligible patients through a specific healthcare law, ensuring that treatment accessibility is not a limiting factor in this population. Thus, our study does not disproportionately favor patients with more complex disease presentations.
Concerning the generalization of the genetic findings, the population of the capital city of Chile is representative of the admixed genetic background of our country and, indeed of several regions of Latin America. Generally speaking, the population of South America originated from the admixture of European settlers and native Amerindians. Thus, even in the face of regional variations in the percentage of the European component, as well as the African contribution and the specific intracontinental native variants, all regions share a relatively common background, where the study of one region can illuminate that of the continent. Thus, we contest the idea that our results are not generalizable. Although we acknowledge that our sample is limited in size and composition, our results provide the basis for future, more encompassing studies. Furthermore, given the scarcity of data in Latin American countries, we firmly believe in the value of this effort.
- No imputation of missing data has been performed, despite certain variables exhibiting over 30% missingness.
Thank you for your observation. In our analysis, variables with more than 20% missing values were excluded. For the remaining variables, mean imputation was applied to continuous variables, while mode imputation was used for categorical variables. Additionally, as now detailed in the Methods section. Lines 702-706
- The genetic association tests do not include any correction for multiple comparisons (e.g. Bonferroni). Given that more than one hundred tests are conducted, statistical significance would only be reached for p-values in the order of 10-4, meaning that some results with p-values merely below 0.05/0.01 may not actually be really significant.
Thank you for your valuable feedback regarding multiple comparison corrections. We acknowledge that conducting more than one hundred tests necessitates stringent significance thresholds. However, a key limitation of our study is the sample size, which affects the statistical power of our genetic association analyses.
We initially applied Bonferroni correction, but the results did not reach significance under this method. Nonetheless, the genetic variants examined in our study have been previously reported as significant in larger IBD cohorts, suggesting their potential relevance despite the limitations imposed by sample size constraints.
Given these considerations, we present our findings as exploratory, emphasizing the need for further validation in larger, independent populations. We will include this information in the discussion section.
We added a commentary in the discussion section lines 550-556
- The ethnicity studied differs from that of the reference GWAS used for the polygenic risk score, which introduces a transferability bias due to the lack of recalibration of the score for the Chilean sample under investigation.
Thank you for your feedback on our study. We appreciate the opportunity to clarify the challenges surrounding the transferability of polygenic risk scores (PRS) when applied to the Chilean population.
As highlighted, our study acknowledges that the ethnicity studied differs from that of the reference GWAS, which introduces a transferability bias due to the lack of recalibration of the score for the Chilean sample. This limitation arises from the absence of genetic studies specific to IBD in Latin American populations. Without a reference dataset tailored to this ancestry, the beta coefficients required for precise PRS calculations are lacking, leading to potential inaccuracies in risk estimation.
While cross-ancestry PRS models have demonstrated varying degrees of applicability, studies indicate that European-derived PRS often lose predictive power in admixed populations, including Latin American groups. The absence of recalibration within our study underscores the broader need for dedicated genomic research within this population to enhance PRS accuracy and clinical utility.
In future work, incorporating admixture-aware statistical methods, leveraging multi-ancestry meta-analyses, and conducting local genome-wide studies may help mitigate these biases.
We added a paragraph in the discussion section about this aspect. Lines 574-586
Minor considerations:
- I recommend introducing in the introduction section some novel pathogenetic mechanisms linking IBD with EIMs. There is emerging evidence regarding the role of the melanocortin system in IBD (https://pubmed.ncbi.nlm.nih.gov/38577176/), which is also implicated in conditions such as arthritis (https://pubmed.ncbi.nlm.nih.gov/37325641/). A brief mention of both studies as relevant references would be appropriate.
Thank you for the suggestion. We have now incorporated the melanocortin system into the introduction as a novel pathogenic mechanism linking IBD and EIMs, ensuring the inclusion of the mentioned references. Lines 62-81
- The quality of Figure 1 is poor; it would be advisable to recreate it using more specialised software (e.g. GraphPad). Additionally, some of the pie chart segments at the bottom (e.g. purple and brown) overlap, obscuring the percentage labels.
Thanks so much we performed the replace of the figure for another.
- In Figures 5 and 6, the use of variable names as they appear in the statistical software is aesthetically unappealing. Labels such as “esr” or “age_diagnoses” should be replaced with properly formatted variable names.
Thanks so much we performed the replace of the figure for another.
Finally, I believe that greater emphasis should be placed, within the discussion, on the translational potential of these findings, on the research pathway that must continue to be pursued in this field, and on what is still lacking for the clinical integration of machine learning-based predictability of EIMs.
Despite these issues, some findings retain a degree of novelty. However, the major limitations of the study lie in its mathematical and methodological approach. These should either be addressed if possible, or clearly and explicitly stated in the limitations section, with a careful recalibration of the strength of the conclusions accordingly.

Round 2
Reviewer 2 Report
Comments and Suggestions for Authors
Please place the title of Figure 1 below the figure and the title of Tables 5 and 6 above each table.
Reviewer 3 Report
Comments and Suggestions for Authors
The authors have provided a detailed point-by-point response. Greater attention has been devoted in the manuscript to the limitations of this work. I therefore believe it can be accepted as an exploratory study.